# Development and Application of a New Flame-Retardant Adhesive

**DOI:** 10.3390/polym12092007

**Published:** 2020-09-03

**Authors:** Xianqing Xiong, Yiting Niu, Zhuorong Zhou, Jie Ren

**Affiliations:** 1Co-Innovation Center of Efficient Processing and Utilization of Forest Resources, Nanjing Forestry University, Nanjing 210037, China; 2College of Furnishings and Industrial Design, Nanjing Forestry University, Nanjing 210037, China; nbbniubaobao123@163.com (Y.N.); zhouzhuorong2020@163.com (Z.Z.); renjie19991205@163.com (J.R.)

**Keywords:** flame-retardant ecological board, impregnated film paper veneer blockboard, flame retardant properties, flame-retardant adhesive

## Abstract

A new design adhesive mixed with flame retardant was developed by an optimized and modified dedicated flame retardant and selected at a suitable proportion between the adhesive and flame retardant as well as the coating amount of the adhesive. The new design adhesive was applied to ecological board production, and the flame-retardant properties of products were examined. The dipping and peeling properties, surface bonding strength, and formaldehyde emission reached the national standard GB/T 34722-2017, the flame retardancy meets the requirements of GB/T 8626-2017, GB/T 20284-2006, GB/T20285-2006, and it also reaches the B1-C level (the nonflammable level in the flame retardant level). This study not only has theoretical guidance but also has strong practical value to provide a basis and data support for the research and development of flame-retardant ecological boards.

## 1. Introduction

According to the development of society and economic progress, people have higher requirements and attention on the safety and comfort functions of building and decorative panels. Ordinary construction and decoration panels are far from meeting the needs of society [1,2]. The scientists and manufacturers must follow the trend of the times and research and develop building decoration materials that meet people’s requirements. Flame-retardant sheets have been widely used in various fields as a functional protective sheet [3,4,5]. In the 1980s, a report of adhesive mixed with flame-retardant materials was found at an international conference [6,7], but no formal research papers have been published. Recently, Back et al. tried to synthesize phosphorus-containing polyol and add it to epoxy adhesive to study its flame retardancy and obtained a relatively high flame-retardant effect [8]. In China, with the development of national laws and standards on flame-retardant materials and the continuous improvement of implementation, especially the implementation of the Ministry of Public Security GB/50222-2017 “Code for Fire Protection of Interior Design of Buildings” [9,10,11], it can promote the more comprehensive application of flame-retardant panels. At present, the research on the materials of construction and decoration panels has become a new hot spot in China.

Ecological board can also be called paint-free board, which is a type of wood-based panel with plywood or blockboard as the substrate and impregnated film as the facing material. Among them, as a kind of eco-board, the surface of impregnated film paper veneer blockboard does not need surface finishing with paint, which is an environmentally friendly material [12]. It has the characteristics of high temperature resistance, acid and alkali resistance, and moisture resistance, and the demand for home improvement is increasing [13,14]. Compared with other types of wood-based panel, it is loved by consumers because the surface does not change its color and peel, and it is easy to process, economical, and environmentally friendly. As the occupied part of indoor space and part of the necessities of life, wooden materials of home improvement are most in contact with the human body in daily life. In addition to meeting the requirements of appearance and comfort performance, the characteristics of safety and environmental protection are directly related to the health of users. Therefore, the formaldehyde emission and flame-retardant effect have become one of the main criteria for the selection when choosing the panels of home improvement. However, currently, the solid wood flame-retardant plates on the market are mainly flame-retardant plywood and flame-retardant blockboard. There is no real flame-retardant ecological board in the industry [15,16]. If the flame-retardant treatment is carried out in the wood board with impregnated film and then used as home decoration material, it can not only effectively solve the indoor fire problem, but it can also increase the customer’s functional requirements for new materials [17,18].

At present, several studies are related to the flame-retardant properties of wooden materials. Wang et al. conducted a comprehensive analysis of the fire resistance of a wooden-structure building by summarizing the related literature and studies. It studied the relationship between the charring rate and fireproof performance [19]. Among them, the material properties (density, moisture content, chemical composition, external dimensions, etc.) and external environment (temperature and oxygen concentration, etc.) have an obvious effect on the charring rate of wood material. Li et al. conducted a combustion performance test by adding five different flame retardants to particleboard with two different raw materials. The results show that the type and amount of flame retardant affect the physical properties of particleboard significantly. However, excess flame retardant will not enhance the flame-retardant performance, but it will reduce the strength between wood shavings to some extent [20]. Hashim et al. tested the dimensional stability and flame-retardant performance by adding three different flame retardants, include sodium aluminate, zinc borate, and aluminum trihydrate in the manufacturing process of medium-density fiberboard. The results showed that the greater the amount of flame retardant, the lower the water-absorption expansion. However, the internal bonding strength decreased obviously. Among the flame retardant, the thermal degradation performance of sodium aluminate was the best [21]. Roger Pedieu et al. made a flame-retardant treatment for particleboard with three different concentrations of boric acid. The study found that the higher the concentrations of boric acid, the less the weight and the slower the spread speed of combustion. Using the white birch inner bark particles as the dispersant can lower the weight loss and ensure the strength of particleboard [22]. According to analysis and summary, we can find that the study on the flame-retardant properties of wood materials mainly focus on the timber buildings and the flame-retardant mechanisms of particleboard and plywood, while there is less study of the technology of ecological board.

According to the material, flame retardant, and addition method, there are three research directions of flame-retardant technology of wood materials, which include the flame-retardant mechanism of wood itself, the flame-retardant, and the treating technology of the flame retardant. The traditional flame-retardant blockboard is to achieve the flame-retardant effect through the process of veneer impregnating the flame retardant. Although the flame-retardant treatment method achieves the flame-retardant effect; however, there are some problems regarding the large amount of materials, high coat of initial investment, complicated manufacturing process, pollution of the environment, moisture absorption deformation, and low strength of the finished board [23], which make it difficult to apply directly to the flame retardant process of the ecological board [24]. It is necessary to innovate the manufacturing process of the ecological board.

To solve these process problems, Nanjing Forestry University cooperated with Zhejiang Shenghua Yunfeng New Material Co., Ltd. to conduct research and development on flame-retardant ecological board products. After a long-term study of flame-retardant treatment technology of the ecological board, we found that the new flame-retardant adhesive synthesized by adding a certain proportion of special flame retardant to the coating adhesive layer has good adhesive performance and an obvious flame retardant effect. This study includes screening and modifying the flame retardant, adjusting the proportion of adhesive and flame retardant, and optimizing the process flow of the flame-retardant ecological board. The flame-retardant properties were measured, and the results obtained were analyzed and discussed in depth. These studies not only expand the application scope of the eco-board, but also provided new ideas and data support for the research and development of a flame-retardant eco-board.

## 2. Materials and Methods

### 2.1. Materials

(1)Core board: the tree species is Chinese fir purchased from Kunhong Wood Industry Co., Ltd., Chufang, China. The specifications are 1260 mm × 2480 mm, the thickness is 12 mm, and the average moisture content is 10%.(2)Middle board (balance layer): the tree species is poplar purchased from Kunhong Wood Industry Co., Ltd., Chufang, China. The size is 1260 mm × 830 mm, the thickness is 2.8 mm, and the average moisture content is 12%.(3)Veneer board (buffer layer): the tree species is poplar purchased from Boyi Wood Industry Co., Ltd., Dezhou, China. The specification size is 1260 mm × 2480 mm, the thickness is 0.7 mm, and the average moisture content is 10%.(4)Flame retardant: a mixture of magnesium hydroxide and melamine phosphate (the weight ratio of magnesium hydroxide to melamine phosphate is 3:1), and the materials were purchased from Kaimei Chemical Technology Co., Ltd., Nantong, China.(5)Adhesive: melamine-modified urea–formaldehyde resin MUF, made by the authors themselves. The pH is 7.5–8.0, the solid content is over 52%, the viscosity is 70–150 mPa.s, and free formaldehyde ≤0.04% [25].(6)Wheat flour: the appearance is white flour solid, the ash content is 0.8%, and the gluten content is 29.8%. The materials were purchased from Yichen Chemical Technology Co., Ltd., Suzhou, China.(7)Impregnated film: melamine impregnated film paper was purchased from Xingmei Decoration Materials Co., Ltd., Linyi, China, the basis weight is 85 g/m^2^, the resin content is 130% and the pre-curing degree is 50%.

### 2.2. Experimental Equipment and Instruments

(1)Experimental equipment: coating machine (Mingshuang Machinery Equipment Co., Ltd., Qufu, China), cold press (Xu Taichang Machinery Co., Ltd., Qingdao, China), hot press (Hongshen CNC Machinery Co., Ltd., Guangzhou, China), and sewing machine (Heibei Songli Machinery Co., Ltd., Xingtai, China).(2)Test instruments: single-combustion experiment device (SBI-1, Nanjing Shangyuan Analytical Instrument Co., Ltd., Nanjing, China). Cone calorimeter combustion device (FCK-1, Nanjing Shangyuan Analytical Instrument Co., Ltd., Nanjing, China).

### 2.3. Experimental Principles and Methods

#### 2.3.1. Screening and Modification of Flame Retardants

Wood materials are flammable materials; however, they have a certain degree of flame retardant, the main reason is that a charring layer can be formed in the combustion process of wood materials. The charring layer can reduce the thermal conductivity, thereby reducing the combustion rate and prolonging the combustion time to some extent. The flame-retardant principle of flame retardant is to prolong the time of wood pyrolysis in order to reduce the mass loss by increasing the amount of charring layer and non-combustible substances [26]. At present, there are three types of flame retardant in the market: organic flame retardant, inorganic flame retardant, and resinous flame retardant [27]. Among them, boron, phosphorus and nitrogen are the most widely used inorganic flame retardants. Based on this, combined with the manufacturing process of flame retardant wood-based panels, this study added the above flame retardant in urea–formaldehyde resin power directly, the process is simple, and the cost is low [28]. However, because the number of glue layers of the blockboard substrate is less than that of plywood, the flame-retardant performance cannot reach the B1-C level. Therefore, it was needed to modify and optimize during the experiment to adapt to the production and processing of wood board with impregnated film paper veneer.

The screening and modification methods are divided into three steps. The first step was to select the flame retardant. Three kinds of urea adhesives produced by three different manufacturers in the market, namely, ZBS, BL, and FQ, were selected according to the production conditions, odors, and costs.

The second step was to pre-test the blending of flame retardant and MUF. The ratio was melamine modified urea–formaldehyde resin (MUF)/flame retardant/wheat flour = 100:25:30.

The third step was to pre-test the flame retardant. Several prepared three-layer eucalyptus plywoods used the new design adhesive prepared in the second step with a 200 g/m^2^ coating amount. The bonding strength was measured and compared with ordinary plywood without flame retardant. The flame resistance mean combustion value was measured in the vertical fire mode for 1 min. The results were the selection of flame retardants for flame-retardant ecological boards.

#### 2.3.2. Preparation of New Design Adhesive

To ensure that the final product reaches the flame-retardant performance index of B1-C level, the selected flame retardant was mixed with MUF to obtain the new design adhesive for ecological boards. The new design adhesive was applied first between the core and the middle plate (primary coating) and then between the middle plate and the single panel (secondary coating) of the ecological board. During the test, the quality ratio of MUF and wheat flour was consistent with the existing production, that is, MUF/wheat flour = 100:30. Focusing on the two core factors of flame retardant and coating amount, an orthogonal test was carried out in two times with coating adhesives, and the design factors are summarized in Table 1.

#### 2.3.3. Process of Flame-Retardant Ecological Board

The core board of blockboard is mainly made of fir in this study, as shown in Figure 1. Considering the structural characteristics of the ecological board such as the front-impregnated film paper layer, front veneer layer, flame-retardant adhesive layer, front mid-layer, flame-retardant adhesive layer, core layer, flame-retardant adhesive layer, back mid-layer, flame-retardant adhesive layer, the back wood veneer layer, the back-impregnated adhesive film paper layer, and combined with the basic structure of common ecological boards, the authors designed the process as shown in Figure 2, and the relevant parameters in the process of pressing are shown in Table 2.

The core was first coated by the new design adhesive and putting the middle plate in place, which was followed by pre-pressing, hot pressing, repairing, maintaining, and sanding to obtain the first coating sample. Subsequently, the first coating sample was coated with a new design adhesive for the second coating and billeted, which was followed by pre-pressing, hot pressing, repairing, maintaining, and sanding to obtain the second coating sample of the flame-retardant ecological board substrate.

#### 2.3.4. Performance Test of Flame-Retardant Ecological Board

(1)Formaldehyde emission: According to the industrial standard and method for quality inspection Q/YFL 0030-2018 “formaldehyde release limit in wood board and its products”, the desiccator method was used for specimens for three parallel determination, and the average value was taken.(2)Dipping, peeling, and surface bonding strength: According to the Chinese national standard GB34722-2017 “impregnated film paper veneer plywood and joinery board”, the specimens for three parallel determination were identified, and the average value was taken.(3)Effect of the hot-pressed board surface: According to the Chinese national standard GB34722-2017 “impregnated film paper veneer plywood and joinery board”, the flame retardant penetration on the surface of specimens after hot-pressing was judged, and the results included two conditions: normal and transparent.(4)Combustion growth rate index and total heat release of 600 s: According to the Chinese national standard GB/T 20284-2006 “monomer of building materials or products combustion test”, the specimens for 3 parallel determination were identified, and the average value was taken.

## 3. Test Results and Analysis

### 3.1. Screening Results of Flame Retardants

According to the flame-retardant screening method in Section 2.3.1, the different characteristics and flame-retardant properties of ZBS, BL, and FQ were obtained and summarized in Table 3.

It can be found from Table 3 that ZBS was the best one compared with the cost, applicability, and performance with MUF, and it was selected to use in this study. The main indicators were that it is a white solid powder with over 36.2% of phosphorus content and over 10.5% of nitrogen content. The used MUF indicators were a pH 7.5–8.0, solid content ≥52%, viscosity 70–150 mPa.s, and free formaldehyde ≤0.04%, respectively.

### 3.2. Analysis Results of Adhesives and Flame Retardants

The performance test and range analysis results of the model are summarized in Table 4 and Table 5, respectively. From the results of Table 4 and Table 5, it can be observed that the two factors have the effects on formaldehyde emission and dip stripping as B > A. A and B are basically the same in terms of the impact on the combustion growth rate index and the total heat release of 600 s. In the first coating, with the increase of the flame retardant ratio (A) and the coating amount (B), both the combustion growth rate index and the total heat release value of 600 s decreased. However, when the flame retardant ratio (A) exceeds 25% and the coating amount (B) exceeds 250 g/m^2^, the combustion value basically almost does not change. When the coating amount (B) is 200 g/m^2^, the dipping and peeling performance was not up to the standard. As the coating amount increases, the dipping and peeling performance gradually improved, and the formaldehyde emission starts to increase.

Considering various indicators and cost, A2B2 was selected as the mixing ratio and coating quantity parameter of the primary coating. Although the formaldehyde emission amount and the dipping and peeling performance of the primary substrate can meet the standard requirements, the flame-retardant performance does not reach the standard, but the value was close to the standard requirements. Based on this result, a layer of flame-retardant adhesive was added to the secondary coating adhesive according to the orthogonal experiment A2B2 plan to improve the flame-retardant performance to meet B1-C requirements. For secondary coating, the performance test result and range analysis of the model are summarized in Table 6 and Table 7, respectively.

It can be seen from Table 6 and Table 7 that the two factors have the effects on formaldehyde emission and surface bonding strength as D > C. C and D were basically the same in terms of the influence on the combustion growth rate index and the total heat release of 600 s. In the secondary coating, with the increase of the flame-retardant ratio (C) and the coating amount (D), both the combustion growth rate index and the total heat release value of 600 s decreased. When the flame retardant ratio (C) exceeds 25% and the coating amount (D) exceeds 250 g/m^2^, the combustion value can reach the B1-C standard requirements. At the same time, when the coating amount (D) is 250 g/m^2^, the surface bonding strength has just reached the standard. As the coating amount increases, the surface bonding strength gradually increases; subsequently, the amount of formaldehyde released also began to increase. When the coating amount (D) exceeded 350 g/m^2^, the board surface effect appeared to penetrate through. Considering various indicators and cost considerations, C2D2 is preferred as the mixing ratio and coating amount parameters of the secondary coating adhesive. By consulting the literature, we can see that Deng et al. made the same conclusion in the study of flame-retardant properties of particleboard after treating the particleboard by flame retardant with different concentrations. It was found that when the content of flame retardant was less than 9%, the flame-retardant performance and bonding strength were gradually improved as the content increased. However, when the content of flame retardant was greater than 12%, the mechanical strength of particleboard was significantly reduced [29]. Luo et al. studied the effects of the time and temperature of pressing and the content of flame retardant in the manufacturing process of particleboard on bonding strength, dimensional stability, and flame-retardant properties. The results showed that within the optimal pressing temperature and time, the longer the time, the better the mechanical strength and dimensional stability. When the flame-retardant content reached 12%, the fireproof performance had been significantly improved; however, the rupture modulus of the particleboard decreased [30].

Therefore, the preferred mixing ratio and coating amount were designed as, in the first coating, the ratio of the flame-retardant adhesive layer between the core and the middle plate was 100 parts MUF, 25 parts flame retardant, 30 parts wheat flour, and the coating amount was 250 g/m^2^. In the second coating, the ratio of the flame-retardant adhesive layer between the core and the middle plate was 100 parts MUF, 30 parts flame retardant, 30 parts wheat flour, and the coating amount is 300 g/m^2^.

### 3.3. Test Results of Flame-Retardant Performance

The flame-retardant ecological board developed in this study was examined by the Dujiangyan National Fireproof Building Material Quality Supervision and Inspection Center. The experiments results were compared with the standard requirements of the flammability of the GB/T 8626-2017 when the flame tip height within 60s and burning dripping matter ignites filter paper phenomenon reaches C-level, the standard requirements of the single combustion performance of the GB/T 20284-2006 when the combustion growth rate index, 600s total heat release, flame spread horizontally reaches C-level, the smoke gas generation rate index and 600s total flue gas generation reaches S2-level, and the combustion drips/particulate reached d0-level, the standard requirements when the toxicity level of the GB/T 20285-2006 reaches t1-level. The results are summarized in Table 8.

It can be seen from Table 8 that the developed flame-retardant ecological board that used the new design adhesive has reached the national standard flame retardant performance B1-C level. It showed that the new design flame-retardant adhesive is suitable for impregnated film paper veneer blackboard with the better performance.

## 4. Conclusions

A new design adhesive was developed by optimizing and modifying flame retardants for adhesives according to the quality requirements of decorative panels for household use. The proportion of adhesives and flame retardants, and the coating amount were optimized by an orthogonal test, which made the eco-board flame-retardant adhesive achieve the flame-retardant performance of flame-retardant impregnated film-coated wood board at the level of B1-C for practical production. All the dipping, peeling, surface bonding strength, and formaldehyde release reached the requirements of national standard and enterprise standard. Based on these results, the industrial production line of flame-retardant impregnated film paper veneer board, or alternatively called flame-retardant ecological board, was established. In the first coating, the most suitable ratio was, MUF/flame retardant/wheat flour = 100:25:30, and the coating amount of the adhesive was 250 g/m^2^. In the second coating, the most suitable ratio was MUF/flame retardant/wheat flour = 100:30:30, and the coating amount the adhesive was 300 g/m^2^, which opened the application of flame-retardant adhesives on ecological boards. Compared with the traditional process of first immersing the veneer of wood-based panels with flame retardant and then gluing, the improved flame-retardant treatment technology can simultaneously carry out flame-retardant and gluing treatment. The advantages of this method are as follows. The amount of flame retardant was reduced to the greatest extent, the release of harmful substance was less, and the possibility of environmental pollution is minimized. Diluting the adhesive layer to a certain extent, the reduction of the adhesive content resulted in a great reduction in the amount of formaldehyde released, which truly achieved low cost, simple process, easy production, and environmental protection.

## Figures and Tables

**Figure 1 polymers-12-02007-f001:**
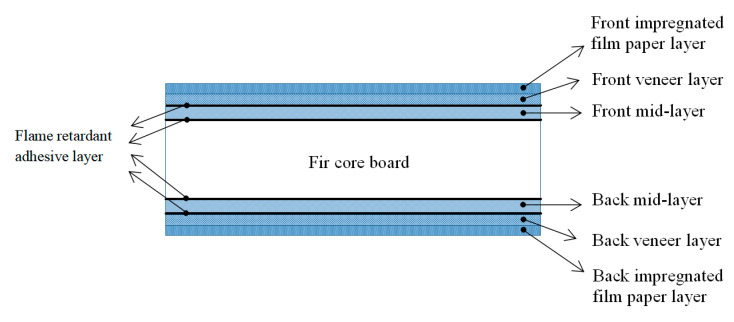
Structure diagram of impregnated film paper veneer blockboard.

**Figure 2 polymers-12-02007-f002:**
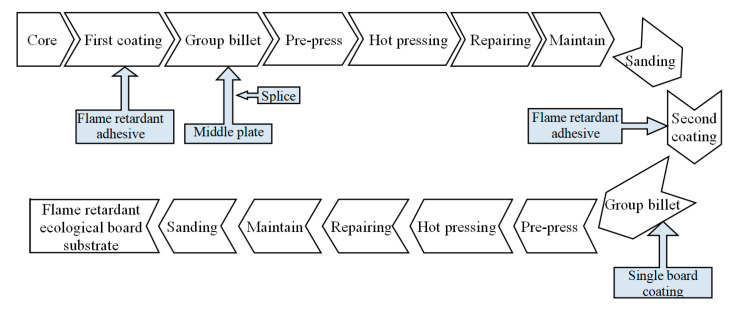
Production process of flame-retardant ecological board.

**Table 1 polymers-12-02007-t001:** Testing factors of the new design adhesive mixed with flame retardant test.

Experiment Procedure	Variable Factors	Level			Judgment Index
		1	2	3	
First coating	Flame retardant (parts) A	20	25	30	Dipping stripping, combustion growth rate index, total heat release amount of 600 s, surface effect of hot pressing plate, formaldehyde release amount
	Coating retardant (g/m^2^) B	200	250	300	
Second coating	Flame retardant (parts) C	25	30	35	Surface bonding strength, combustion growth rate index, total heat release of 600 s, surface effect of hot pressing plate, formaldehyde release
	Coating retardant (g/m^2^) D	250	300	350	

**Table 2 polymers-12-02007-t002:** The parameters in the process of pressing.

Experiment Procedure	Pre-Pressing		Hot-Pressing		
	Pressure (MPa)	Time (h)	Temperature (°C)	Pressure (MPa)	Time (min)
First coating	0.6	1	125	0.6	16
Second coating	0.6	0.5	120	0.6	12
Veneering	—		127	0.7	7

**Table 3 polymers-12-02007-t003:** Characteristics of different flame retardants and their effects after being formulated with adhesives. MUF: melamine-modified urea-formaldehyde resin.

Flame Retardant	Original Characteristics			The Effect after Mixing with MUF	Determination of Flame Resistance		
	Status	Odor	Cost (Yen/t)		Strength (MPa)	Whether There is an Open Flame	Whether Burning Dripping Occurs
ZBS	White powder	Odorless	9200	Good glue fluidity with a few particles	1.1	No	No
BL	White powder	Odorless	15,000	Thick glue without fluidity	0.7	No	No
FQ	White powder	Odorless	6500	Good glue fluidity with a few particles	0.9	There is a short flame	Yes
Ordinary plywood					1.2	Burned	Yes

**Table 4 polymers-12-02007-t004:** Test results of one-time sizing optimization test.

Sample Number	Factor and Level	Dipping and Peeling	Combustion Growth Rate Index (W/s)	600 s Total Heat Release (MJ)	Formaldehyde Emission (mg/L)	Hot Plate Surface Effect
1	A1B1	NG	283	18.2	0.2	OK
2	A1B2	OK	274	17.6	0.3	OK
3	A1B3	OK	272	17.3	0.4	OK
4	A2B1	NG	274	17.6	0.2	OK
5	A2B2	OK	264	17.2	0.3	OK
6	A2B3	OK	263	17.2	0.3	OK
7	A3B1	NG	273	17.7	0.2	OK
8	A3B2	OK	262	17.0	0.3	OK
9	A3B3	OK	263	17.0	0.4	OK

**Table 5 polymers-12-02007-t005:** Range analysis of one-time sizing optimization test.

Performance Index	Factor	Average 1	Average 2	Average 3	Range
Combustion growth rate index (w/s)	A	276	267	266	10
	B	277	267	266	11
600 s Total heat release (MJ)	A	17.7	17.3	17.2	0.5
	B	17.8	17.3	17.2	0.6
Formaldehyde emission (mg/L)	A	0.3	0.27	0.3	0.03
	B	0.2	0.3	0.37	0.17

**Table 6 polymers-12-02007-t006:** Test results of two-time sizing optimization test.

Sample Number	Factor and Level	Surface Bonding Strength (MPa)	Combustion Growth Rate Index (W/s)	600 s Total Heat Release (MJ)	Formaldehyde Emission (mg/L)	Hot Plate Surface Effect
1	C1D1	0.6	246	16.4	0.2	OK
2	C1D2	0.8	238	15.3	0.3	OK
3	C1D3	1.0	234	15.2	0.4	Infiltration
4	C2D1	0.7	240	15.2	0.2	OK
5	C2D2	0.8	230	14.2	0.2	OK
6	C2D3	0.9	230	14.3	0.4	Infiltration
7	C3D1	0.6	235	15.0	0.2	OK
8	C3D2	0.8	229	14.2	0.3	OK
9	C3D3	1.1	228	14.0	0.3	Infiltration

**Table 7 polymers-12-02007-t007:** Range analysis of two-time sizing optimization test.

Performance	Factor	Average 1	Average 2	Average 3	Range
Combustion growth rate index (w/s)	C	239	233	230	9
	D	240	232	231	9
600 s Total heat release (MJ)	C	15.6	14.6	14.4	1.2
	D	15.5	14.6	14.5	1
Formaldehyde emission (mg/L)	C	0.33	0.27	0.27	0.06
	D	0.2	0.27	0.37	0.17
Surface bonding strength (MPa)	C	0.8	0.8	0.83	0.03
	D	0.63	0.8	1	0.37

**Table 8 polymers-12-02007-t008:** Performance of joinery board decorated by flame-retardant impregnated thermosetting resins paper.

No.	Test Items	Test Methods	Standard Requirement			Result	Conclusion
1	Flammability	Flame tip height within 60 s, mm	GB/T 8626-2017	C-level	≤150	55	OK
		Burning dripping matter ignites filter paper			The filter paper is not ignited	The filter paper is not ignited	
2	Single combustion performance	Combustion growth rate index (W/s)	GB/T 20284-2006		≤250	227	OK
		600 s Total heat release (MJ)			≤15	14.0	
		Flame spread horizontally			Did not reach the sample long edge	Did not reach the sample long edge	
		Smoke gas generation rate index (m^2^/s^2^)		S2-level	≤180	14	OK
		600 s Total flue gas generation (m^2^)			≤200	67	
		Combustion drips/particulates		d0-level	No burning drips/particles within 600 s	No burning drips/particles within 600 s	OK
3	Toxicity level		GB/T 20285-2006	t1-level	ZA3-level	ZA3-level	OK
Combustion growth rate index, FIGRA0.2MJ = 227 w/s							

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
