# Peer review of "Development and Application of a New Flame-Retardant Adhesive"

_polymers, 2020, doi:10.3390/polym12092007_

Round 1

Reviewer 1 Report

I found your paper interesting. I have some comments and suggestions that I strongly believe that will improve its quality.

Comments and suggestions:

1) Introduction – is adequate. Unfortunately, the flame retardant technology of wood materials is focused mostly in Chinese references. Many European references which are related to the flame retardant technology of plywood, particleboard and other wooden materials are missing.

2) Throughout the text, the authors use the terms "eco-board" or "ecological board". What is the justification for the studied blockboard to belong to the "ecological board"? 

3) The authors use non-formaldehyde flame-retardant adhesive and talk about ecological board and on other side determine the emission of formaldehyde. Why? 

The Materials and Methods are not adequately described. I would very much like to see the following information to be added:

4) It is expedient to present the scheme of a design of a blockboard with designation of all corresponding layers. 

5) Core, middle and veneer boards have different moisture content: 10, 12 and 10%, respectively. How to explain the different moisture content of the blockboard's layers? How will the difference in the moisture content of the layers affect the development of stresses in the pressing process of the blockboard and its strength? 

6) The detailed information about MUF adhesive is needed, for example viscosity, solid content, pH, curing time, and formaldehyde content. This information is missing in the text. 

7) chapter 2.3.1., lines 117-130 belong rather to the Introduction section.

8) What are the properties of the new adhesive system: viscosity, solid content, curing time, pH, and formaldehyde content? This information is missing in the text. 

9) The most readers are unfamiliar with Chinese standards. Please describe in detail the methods of the determination: the combustion growth rate index, the total heat release of 600s, the effect of the hot-pressed board surface, the formaldehyde release amount, and the surface bonding strength. 

10) There is no information on what method the adhesive was applied to the glued surfaces, what were the pressing modes (pressure, temperature, time) of the blockboard. 

11) How were core and middle boards made? Were they purchased or made in a laboratory? What adhesives are used to make them? 

12) line 184: free formaldehyde ≤0.04? unit of measurement?

13) line 133: Information on the properties (viscosity, solid content, curing time, pH, formaldehyde content) of ZBS, BL and FQ adhesives is omitted. This complicates the analysis of the results. 

14) What was the weight loss of the samples during the fire resistance tests? This parameter is very important and should be added to the text.

15) Specify “surface bonding strength” and how was it determined? What was the adhesive strength between core board and first coating layer, between core and middle boards, between middle and veneer boards, between veneer board and impregnation film paper....? 

16) The Materials section does not consist anything about “the front and back impregnated film paper layers”. Please provide a detailed description of the impregnated film. 

17) The thickness of the flame-retardant adhesive layer is extremely small compared to the thickness of the core and middle boards or total thickness of blockboard. The edges of the boards are open, unprotected. Therefore, the fire will spread over these layers and the flame-retardant adhesive layer will not be able to protect them as well as blockboard itself. Do you believe that will be enough for the manufacture of fire-resistant blockboard to use only flame-retardant adhesive to bond layers of the board without protecting the layers of the board? 

Chapter 3. Test Results and Analysis.

18) This chapter should be modified since it does not contain any references, there is no deep discussion of the obtained results. Therefore, more attention should be paid to the deep discussion of the findings, comparing them with other known works in the field of fire resistance of wood materials, the disclosure of the mechanism for increasing the fire resistance of the blockboard.

Conclusions

19) lines 257-260: "The manufacturing process was simplified, the production time was reduced and the production efficiency was improved. The manufacturing process was optimized to eliminate the process defects caused by the flame retardant soaking the veneer." These paragraphs should be deleted since the authors did not investigate  the production time, the production efficiency and the process defects and moreover did not compare it with traditional process.

Reviewer 2 Report

The paper concerns an interesting topic, which is the use of high resistance glue for boards.
In my opinion, chapter 2, concerning materials and methods used, is completely illegible. It must be rewritten. In addition, I think that the authors, when referring to Chinese standards, should specify what they are similar or different to European ones. Simply stating the number of a standard is not enough, because it requires the reader to look for additional information, which is not always easy to obtain.

Round 2

Reviewer 1 Report

Lines 37-40. Please add references.